# Preliminary Investigation of Schmalhausen’s Law in a Directly Transmitted Pathogen Outbreak System

**DOI:** 10.3390/v15020310

**Published:** 2023-01-22

**Authors:** Antoine Filion, Mekala Sundaram, Patrick R. Stephens

**Affiliations:** Department of Integrative Biology, 501 Life Sciences West, Oklahoma State University, Stillwater, OK 74078, USA

**Keywords:** disease ecology, Schmalhausen’s law, ecological boundaries, spillover, filovirus, disease emergence

## Abstract

The past few decades have been marked by drastic modifications to the landscape by anthropogenic processes, leading to increased variability in the environment. For populations that thrive at their distributional boundaries, these changes can affect them drastically, as Schmalhausen’s law predicts that their dynamics are more likely to be susceptible to environmental variation. Recently, this evolutionary theory has been put to the test in vector-borne disease emergences systems, and has been demonstrated effective in predicting emergence patterns. However, it has yet to be tested in a directly transmitted pathogen. Here, we provide a preliminary test of Schmalhausen’s law using data on Marburg virus outbreaks originating from spillover events. By combining the two important aspects of Schmalhausen’s law, namely climatic anomalies and distance to species distributional edges, we show that Marburgvirus outbreaks may support an aspect of this evolutionary theory, with distance to species distributional edge having a weak influence on outbreak size. However, we failed to demonstrate any effect of climatic anomalies on Marburgvirus outbreaks, arguably related to the lack of importance of these variables in directly transmitted pathogen outbreaks. With increasing zoonotic spillover events occurring from wild species, we highlight the importance of considering ecological variability to better predict emergence patterns.

## 1. Introduction

Anthropogenic changes have greatly altered the delicate balance that dictates species distributions worldwide. By either introducing new species, encroaching on natural habitats, or changing the landscape, human activities have completely modified expected species distributions and interactions in many ecosystems [1]. For species that already live in physiologically stressful conditions in areas of greater variability, for instance at the edge of their distributional range, these modifications can affect them drastically. For instance, many mechanistic processes, such as predation, competition, and others, are often heightened in ecotones [2,3]. Following the same rationale, greater variability in the environment has been demonstrated to increase the chances of disease spillover and subsequent zoonotic outbreaks [4]. For example, mammalian hosts of Lyme disease (*Borrelia burgdorferi*) are more abundant in ecotones and contribute to larger outbreaks in these areas [5]. With the potential for more zoonotic outbreaks, gaining a deeper understanding of the influence of environmental variability in spillover events would prove as a valuable interest for conservation management from a One Health perspective.

Schmalhausen’s law holds that species pushed to the edges of their ecophysiological tolerances in any dimension will express unusual phenotypes, as genotypic expression occurs in conditions that have been rare or absent during species’ evolutionary history [6]. Applied to diseases, Schmalhausen’s law predicts that temporal or spatial boundaries of ecological systems are close to the edge of species physiological tolerances, and therefore susceptible to minor environmental changes potentially leading to unusual outbreak dynamics [7]. The key advantage of working within this evolutionary theory is that it integrates both spatial and temporal variability into a unified conceptual framework to apply toward disease outbreaks. For example, [8] showed that increased variability in both temperature and rainfall was positively related to mosquito emergencies, contributing to outbreaks of the Western Nile Virus in Texas. So far, Schmalhausen’s law has been applied to vector-borne disease systems, such as avian malaria [7] and leishmaniasis [9], presumably because conditions that are close to the physiological limits of either the vectors or the pathogen itself allow for unusual outbreak dynamics with minor environmental changes. There is some evidence that directly transmitted parasite prevalence is higher at their host’s range edge [10]. However, whether Schmalhausen’s law may affect outbreak characteristics remains to be tested in a directly transmitted pathogen.

There are multiple mechanistic ways in which individuals or populations of a host species being pushed to the edge of their ecophysiological tolerances will potentially result in an increased frequency of spillover. For instance, immunity to any given pathogen can be a function of environmental tolerance, in which a harsher environment limits an individual’s resource allocation to the immune system, in turn making them more prone to infection [11,12]. Local landscape features are also known to change immune functions between individuals of the same species [13]. As such, increased variation in immunity at species range edges might lead to increase spillover risk toward other populations. Another mechanistic path that would result in an increased spillover rate is an increased rate of contacts at the human/wildlife interface, particularly in cases when species range edges occur near transitions zones between biomes such as forest edges, resulting in spillover event due to an increase in contact with natural hosts shedding their pathogens [14].

Here we present a preliminary test of Schmalhausen’s law in a directly transmitted zoonotic pathogen, Marburg virus (*Marburgvirus marburgvirus*), a filovirus related to the Ebola genera of viruses Though outbreaks of this pathogen are relatively rare, they also generally show extremely high mortality [15]. For example, the largest outbreak to date showed a case fatality rate of 88% among over 100 human cases [16,17,18], suggesting that Marburg virus could post a significant threat were it to ever spread widely. Thus, there is an urgent need to better understand the mechanisms behind this pathogen’s outbreaks. Marbug virus outbreaks are an ideal system for preliminary analyses of Schmalhausen’s law in a directly transmitted pathogen since it is a relatively simple system, with one reservoir being associated with outbreak events, the Egyptian fruit bat (*Rousettus aegyptiacus*) [19,20]. This allowed us to test Schmalhausen’s law by focusing on geographic range effects in a single well-characterized wild host species, simplifying our analyses considerably. Moreover, it has been demonstrated that roosting Egyptian fruit bats release the Marburg virus in seasonal pulses, suggesting a temporal influence in spillover events [21]. Recent advances in remote sampling [22] have also made it possible to obtain month-to-month climatic anomalies for the locality of every documented Marburg virus outbreak. As such, we feel confident that we are able to incorporate the spatio-temporal conceptual framework of Schmalhausen’s law into a disease dynamic that is extremely important from a One Health perspective.

Arguably, as opposed to vector-borne pathogens, small environmental changes would have far less impact on a non-vector host species: endothermic mammalian hosts in particular are likely far less influenced by the types of minor climatic events that prove impactful to vectors. On the other hand, species at the edge of their distributional range will still be affected by multiple environmental stressors. If Schmalhausen’s law applies to directly transmitted pathogens, we predict that minor climatic anomalies, represented by events outside the normal standard deviation for an area, will affect outbreak size. In addition, we predict that outbreaks at the edge of the Egyptian fruit bat spatial range will be more severe than those in the center of this species’ distribution. If we can detect any evidence of the influence of Schmalhausen’s law in this relatively rare pathogen maintained by a single endothermic reservoir, it is likely that this evolutionary law would apply to other directly transmitted pathogens as well.

## 2. Materials and Methods

### 2.1. Data Collection

All outbreaks included in our study (Figure 1A) were compiled from [23], CDC outbreak data [20], and ProMED [24]. We then pruned the dataset to include only confirmed Marburg virus outbreaks that originated from a wild source (i.e., we excluded all data points from accidental laboratory spillover events). For each outbreak, we compiled the date of the outbreak, its geographical original location, and the number of human cases, allowing us to work with 13 outbreaks ranging from 1975 to 2021, inclusively.

We obtained monthly rainfall temperatures from the Historical monthly weather data [25], downscaled with WorldClim 2.1; [22]. We then extracted monthly rainfall and minimum monthly temperature for each set of coordinates in our database (N = 13, package: raster; [26]. We focused on minimum monthly temperature somewhat ar bitrarily from among several highly correlated measures of temperature we could have focused on (e.g., maximum monthly temperature, mean monthly temperature). However, we speculate that in a tropical environment cold temperature anomalies seem to be particularly stressful to hosts and their pathogens. We focused on this measure of environmental variation rather than historical anomalies derived from timeseries analyses (e.g., [27]) as it closely mirrors the measures of climatic variation used in previous studies that supported the influence of Schmalhausen’s law in vector-borne disease systems [7,8,9].

We obtained the spatial range of the Egyptian fruit bat as a polygon shapefile from the IUCN red list repository [28]. Using this information, we then extracted the minimum distance between each location and the edge of the closest polygon, representing a minimum distance between a Marburg virus outbreak and the distributional edge of the natural reservoir of the Marburg virus using ArcGIS V10.4 [29].

### 2.2. Data Analysis

All analyses were performed using R version 4.0.2 [30]. We provide our R code as Appendix A to allow readers to fully reproduce our results and provide a useful starting point for similar analyses in other systems. A known source of bias that could influence our response variable is human population density [31,32], where larger outbreaks occur in areas of higher density. To ensure that our results were not influenced by this, we downloaded the Population Density Grid, v3 (SEDAC) for the year 2000, and extracted human population density for our geographic coordinates. We then performed a Bayesian regression analysis with a negative binomial distribution (package: brms; [33]), where our response variable was the outbreak size and the predictor was human population density. We found no relationship between these two variables (effect size (slope) = −0.001, lower Credible interval = −0.002, upper Credible interval = 0.001), indicating that our results are unlikely to have been driven by variation in human population density.

First, to understand the importance of our climatic variables on outbreak severity, we calculated the moving average and standard deviation for both rainfall and temperature for each sample site with a window of 30 days (package: tidyquant; [34]). We then randomly subsampled 100 data points for which we knew there were no reported outbreaks, thereby creating a pseudo-absence data frame. Afterward, we calculated the 95% confidence intervals of the standard deviation of environmental variables for our pseudo-absence data frame, and checked if the moving standard deviation for up to a year before outbreaks at each site fell inside it, allowing us to check for anomalies in the time series for each Marburg virus outbreak. We considered it unlikely that the ecophysiological effects of climate on outbreak risk predicting Schmalhausen’s law would exhibit a temporal lag greater than one year. These analyses showed no evidence of any influence of climatic variation on outbreak risk (see results), and so no additional analyses of these variables were conducted.

To determine whether there was a significant overall influence of host species spatial range in the Marburg virus outbreak, we used a Bayesian population-level model (package: brms; [33]) with weakly informative priors set for our only coefficient. We used four chains, with 10,000 iterations per chain (5000 for warmup, 5000 for sampling) We used the human outbreak size (count data) as a response variable with a negative binomial distribution (link function: log) and the closest distance to the edge of the Egyptian fruit bat distributional range as our only population-level effect (see Figure 1B). Lastly, we checked for convergence of our predictor using the RHAT indicator (at convergence, RHAT = 1).

## 3. Results

The moving monthly standard deviation for rainfall did not deviate from the expected 95% confidence intervals, and neither did the monthly standard deviation for temperature. In contrast, we detected a relevant effect from the distance to our target species distributional range on the Marburg virus outbreak, with a mean effect size (slope) of −0.01 (lower Credible interval: −0.02, Upper Credible interval: −0.01). However, keeping in mind that this result may be driven by an outlier in a small dataset, we emphasize caution in the interpretation of this result.

## 4. Discussion

Schmalhausen’s law predicts that minor environmental changes should strongly affect outbreak risks in variable areas, particularly for species living at the edge of their distributional range or environmental tolerances in general [6]. While this prediction has been tested in multiple vector-borne pathogen systems ([7,8,9], there is a clear lack of literature regarding Schmalhausen’s law indirectly transmitted pathogens. In terms of large-scale disease dynamics, Schmalhausen’s law predicts that species near the edges of conditions they can tolerate due either to spatial or temporal variation in environmental conditions have the potential to generate outbreaks with unusual characteristics. Here, we show how the potential influence of Schmalhausen’s law can be tested for both in the context of temporal climatic anomalies and distance to species distributional edge (i.e., to the spatial limits of conditions a species is able to survive and reproduce) using Marburg virus outbreak as our study system. Directly transmitted pathogens, even those similar to the Marburg virus such as Ebolaviruses and hemorrhagic fevers in general [35], tend to have numerous potential wild reservoirs [36]. In contrast, Marburg virus is a relatively simple system with a single primary reservoir [19,20]. This makes it an ideal choice for a “proof of concept” study.

While we failed to detect any direct influence of climatic variability on the Marburg virus outbreak, we still believe that there could be an underlying mechanism dictating the effects of climatic variability on disease outbreaks in the context of Schmalhausen’s law. For instance, the pulse of resource hypothesis would prove a valuable alternative explanation for any climatic patterns related to Schmalhausen’s law, with species gathering in resource-rich environments after a minor climatic disturbance [37] thus enhancing the potential for pathogen spillover.

The main finding of this study is that distance to species distributional edge seems to be a key driver in outbreaks severity of Marburg virus in Africa. Schmalhausen’s law predicts that when species are pushed to the limits of their ecophysiological tolerances such as in the boundaries of an ecosystem [6], minor environmental changes can lead to unusual outbreak dynamics. Here, we show that the Marburg virus outbreak pattern seems to follow this aspect of Schmalhausen’s law, with outbreaks having an increased severity and the edge of the Egyptian fruit bat distribution. However, the mechanisms behind this remain a matter for debate and there are few tests of any of the predictions of Schmalhausen’s law in directly transmitted pathogens (but see [10]). We posit that the pattern observed here could be due to the presence of either natural or artificial ecotones, increasing the rate of contact between the Egyptian fruit bat and human settlements, resulting in a higher probability of a spillover event [4,38]. This phenomenon has been demonstrated in multiple systems, for instance in intensified agricultural landscape (see systematic review by [39], and we believe this could provide a plausible explanation for what we observe here as well. Another equally suitable explanation for our result would be a change in R_0_ of the Marburg virus in stressful areas. For instance, [40] showed that many directly transmitted pathogen transmissions were increased with climatic perturbations, mainly due to changes in habitat range and increased rates of contact between species. Lastly, it is possible that individuals present at the edge of their ecophysiological tolerance, and hence in stressful conditions, will not be able to allocate sufficient resources toward their immunity, resulting in increased chances of spillover in these environments [13]. As such, we bring forward that an increase in transmission rates of the Marburg virus at the edge of the reservoir species distribution would create larger outbreaks.

Of course, the result presented here should be interpreted carefully. The obvious caveat of this study is the small sample size: we decided to use the Marburg virus outbreak to test Schmalhausen’s law in a directly transmitted pathogen since it remains a simple system of critical interest from a One Health perspective. However, it is also a relatively rare pathogen, with only 13 documented natural outbreaks to date (Figure 1A), which limited our statistical power. In addition, the main result of our study may be biased by an outlier, with the only large (100+ cases) outbreak of the Marburg virus happening at the edge of the Egyptian fruit bat range in Angola. This remains an odd, but nevertheless interesting, an observation which would benefit from increased research. Despite these limitations, we were able we show preliminary support for the influence of Schmalhausen’s law on directly transmitted pathogens. A major improvement in this system would be to measure viral prevalence in species edge compared to the center of their distribution, as this help distinguish whether mechanisms related to hosting physiology and immune function or alternative mechanisms such as increased contact rates between the reservoir and other species including human play a role.

The purpose of our study was to determine whether we could detect evidence of the disease dynamics predicted by Schmalhausen’s law in this simple one-reservoir system in which vector-borne transmission does not play a role, as a proof-of-concept. We thus focused on factors expected to push populations of the primary reservoir of the Marburg virus to the edges of their physiological tolerances. Our results demonstrate that distance to the range edges of known hosts may be an important factor to include in future studies of outbreak risk, even for directly transmitted pathogens with endothermic reservoirs. However, previous studies have shown that a number of other factors including non-anomalous climatic variation, patterns of deforestation, and human population density [41,42] are also important predictors of filovirus outbreak risk. In addition to host range effects, these would be important to consider in future studies focused on building the most accurate possible predictive models of outbreak risk of filoviruses or other pathogens.

With an increase in pathogen spillover leading to zoonotic events in many terrestrial systems around the world, we believe that there is a strong need to better understand the drivers of past and current outbreaks in order to mitigate the impact of future pathogen spillover events.

## Figures and Tables

**Figure 1 viruses-15-00310-f001:**
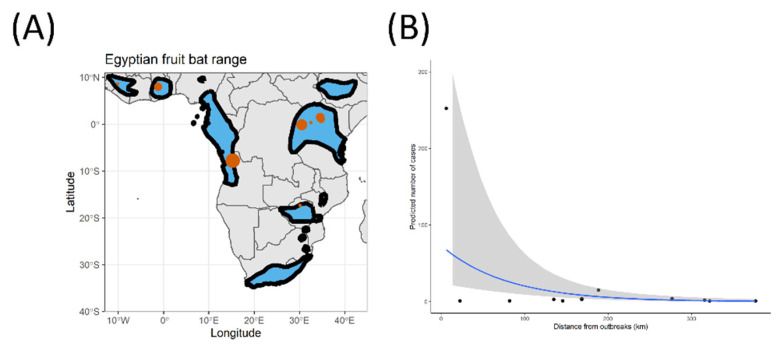
(**A**) Distributional range of the Egyptian fruit bat (*Rousettus aegyptiacus*). Black borders represent the edge of this species’ distribution in each territory. Red dots represent Marburg virus outbreaks in human populations, independent of the year of the outbreak. Note that multiple outbreaks occurred at some of these locations. The size of the red dot represents the size of the largest outbreak that occurred at each location. Range data for the Egyptian fruit bat from IUCN red list repository. (**B**) Predicted risk of Marburg virus infection based on our Bayesian model prediction. Grey polygon represents 95% credible intervals.

## Data Availability

Upon acceptance of this manuscript, dataset and code will be provided via DRYAD repository. The code is not novel code but is there to help other researcher achieve reproducibility with our dataset.

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
