# Peer review of "Preliminary Investigation of Schmalhausen’s Law in a Directly Transmitted Pathogen Outbreak System"

_viruses, 2023, doi:10.3390/v15020310_

Round 1

Reviewer 1 Report

The main purpose of this study is to examine Schmalhausen’s law for Marburg virus outbreaks using climate variables and the spatial distribution of animal reservior.  Although the study concept is interesting, the study design and analysis can be improved.

1. Historical anomaly indices might be a better parameter to evaluate the influences of climate.

2. Human population density might be an important parameter relevant to disease detection because Marburgvirus is a rare disease.   

Author Response

The main purpose of this study is to examine Schmalhausen’s law for Marburg virus outbreaks using climate variables and the spatial distribution of animal reservior.  Although the study concept is interesting, the study design and analysis can be improved.

REPLY: We thank the reviewer for finding our study interesting. Below, we present a detailed answer the to the reviewer’s concerns.

  1. Historical anomaly indices might be a better parameter to evaluate the influences of climate.

The method that we use to define "anomalies" closely mirrors the methods used by previous studies that have tested for the effects of Schmalhausen's law in vector-borne disease systems. While we agree that it would be potentially interesting to examine other definitions of anomalies based on formal timeseries analyses, such as Climdex (Silman et al. 2013), in the highly likely case that these analyses would also fail to show any significant relationships it would be unclear if the difference in our results was related to our study system or due to using different definition of "anomaly" from previous studies of Schmalhausen’s law and climate. We have clarified this point for readers in the methods (see lines 125-129)

  1. Human population density might be an important parameter relevant to disease detection because Marburgvirus is a rare disease.   

REPLY: We agree that human population density would be important to include in our models if our central goal was to model the chances of an outbreak occurring and being detected. However, our primary purpose was to test the specific ecological predictions of Schmalhausen’s law in a relatively simple one reservoir system that lacks vector borne transmission, as a proof of concept study.

That being said, we ran a simple linear model to understand the effect of this predictor had in our database. As we found no significant influence of it, we feel confident that this bias does not play an important role in our dataset. We have added a small paragraph regarding this in the methods (see lines 138-146), and expanded the discussion on this issue (see lines 263-266)

Reviewer 2 Report

In this short commentary, the authors test Schmalhausen’s Law (the idea that animals at the limits of their range are more vulnerable to environmental variation) in the Rousettus aegyptiacus — Marburg virus system. My main comment here is that I think the authors could better explain the link between range limits and spillover, particularly the jump from host occupancy and human exposure. There are also some more minor comments regarding the analyses, particularly predicting MARV across all of continental Africa based on just data points and not restricting their prediction grid to the Egyptian fruit bat distribution (which is limited to only patches of the continent).

L15 and elsewhere: The virus should either be called by its common name (Marburg virus; MARV) or by its scientific name “Marburg marburgvirus”. The current naming conflates the two. 

L45-57: Currently missing from the Introduction is discussion of how hosts being at the margins of their range would affect disease outbreaks in the host or spillover events to novel hosts. There is a large literature in ecoimmunology showing immune impairment at range limits (e.g., Ardia 2005 Ecology, Krama et al. 2013 Journal of Ornithology, Becker et al. 2019 ICB), which posits that susceptibility to infection or intensity of parasite shedding would thus be greatest at the range margins (and therefore that is where humans would have the greatest risk of exposure to infectious hosts, in the spirit of Plowright et al. 2017’s Nature Reviews Microbiology framework on spillover hierarchies). I think the jump from hosts being at the range limits to spillover risk needs to be introduced in a much more mechanistic manner here. 

L54: The disease is mis-spelled here (i.e., “Leischmaniasis” should be “leishmaniasis”).

L57: Although not likely a directly transmitted parasite, the authors should also cite Briers 2003 “Range limits and parasite prevalence in a freshwater snail”, which provides a non-vector example of how parasite prevalence is greatest at the species range. 

L60: See above comments on virus naming; please use the full virus name or just Marburg virus (and then MARV thereafter). 

L60: This statement isn’t really accurate. MARV and EBOV are different genera of viruses. It would be more accurate to say “a filovirus related to the Ebola genera of viruses.”

L68: The authors should also cite Amman et al. 2012 PLoS Pathogens here, which provides a direct test of shedding pulses in Rousettus bats being associated with spillover to humans.

L77-L88: See my above comment, but I think this introduction is really lacking information on the mechanistic jump from host occupancy in a range margin habitat to spillover. I think the authors need to better acknowledge that range limits are unlikely to affect spillover in a direct sense (e.g., reservoir-to-human contact) but rather affect more upstream processes (e.g., immunity, shedding) that then percolate down to spillover risk. In an ideal world, you would test of being at the range limit affects virus prevalence or shedding in the host, although I understand that isn’t possible with the dataset here. But I do think the authors should acknowledge that limitation here.

L106-110: The authors should clarify here that the Rousettus distribution is patchy, such that it is composed of multiple polygons. In addition, is there any way to account for whether outbreaks are observed at the generally more northern or southern limit of the bat range? E.g., you could theoretically have an outbreak in very western Ethiopia that in this framework would correspond to the “edge” of one of the IUCN range polygons, but this is still closer to the core of the bat range than say an outbreak that would occur in South Africa or Turkey (very much at the southern or northern limits of the range).

L111: Please clarify the sample sizes used in the analyses here (i.e., is it N=13?).

L112-125: Since the authors use a regression framework to analyze outbreak size ~ range limit, why do you not do the same for the climatic predictors? It’s unclear why two separate analyses are used to analyze predictor A and predictor B. 

L126: The authors mention using a multi-level model, but no random effects are apparent in the structure stated in L130 (which looks like outbreak size ~ range limit distance). Please also clarify the other Bayesian parameters for the brm() function (how many chains, how many iterations, how much burn in, etc). 

L131: Why was this particular geographic extent chosen? From Figure 1, it is clear that the authors do not predict on the whole Rousettus geographic range, just that within the African continent. Likewise, I do not see merit in predicting across the whole African continent if the Egyptian fruit bat is not distributed across the whole continent. Why not only predict within the known polygons? In general, I’m also pretty wary of predicting spillover risk across an entire continent based on just 13 data points.

Figure 1: A legend should be provided in panel A to indicate outbreak size; can the point size also be increased and/or made transparent? From my understanding of the methods, there should be 13 outbreaks but I can only see 7-8 here. Also note my above comment, as both figures restrict the Egyptian fruit bat geographic range (this should be clarified in the legend). I would suggest using a better color scale for panels A and B (given concerns about red-green colorblindness). It is also very hard to make out the areas of high predicted human cases in panel B. Also see my above comment about predicting across the African continent, given that this bat species is not broadly distributed. 

L146: “Effect size” is a bit vague. Is this not just the posterior mean of the slope for distance to the range limit? It would be more appropriate to describe this as simply the slope/beta; I would also clarify that the CIs are credible intervals from brm(). 

L174-192: See my above comments about the Introduction, the authors should discuss the (likely) possibility that range limits affect bat immunity in ways that facilitate greater intensity of infection and/or virus shedding. I’m not sure the ecotone argument really makes much sense unless there is compelling evidence that ecotones are more likely at the range limits. Since the range limits are shaped by climatic or other biophysical variables, I doubt you’d find range limits are also areas of greater human-wildlife interaction. Either way, I think the idea stress at range limits mentioned here necessitates some discussion of this alternative explanation. 

L202: The authors should also briefly discuss the caveat that many processes need to occur in-between the host being present in a range limit habitat to spillover occurring and that measures of viral prevalence or shedding in such habitats would be important data as well. 

Author Response

In this short commentary, the authors test Schmalhausen’s Law (the idea that animals at the limits of their range are more vulnerable to environmental variation) in the Rousettus aegyptiacus — Marburg virus system. My main comment here is that I think the authors could better explain the link between range limits and spillover, particularly the jump from host occupancy and human exposure. There are also some more minor comments regarding the analyses, particularly predicting MARV across all of continental Africa based on just data points and not restricting their prediction grid to the Egyptian fruit bat distribution (which is limited to only patches of the continent).

REPLY: We agree with the reviewer that the link between range limits and spillover should be better explained. We have now reworded the introduction to expand on this issue (see lines 61-73).

L15 and elsewhere: The virus should either be called by its common name (Marburg virus; MARV) or by its scientific name “Marburg marburgvirus”. The current naming conflates the two. 

REPLY: We have changed naming to Marburg virus in all instances, save in the introduction where we also use the scientific name for the sake of clarity (see line 76).

L45-57: Currently missing from the Introduction is discussion of how hosts being at the margins of their range would affect disease outbreaks in the host or spillover events to novel hosts. There is a large literature in ecoimmunology showing immune impairment at range limits (e.g., Ardia 2005 Ecology, Krama et al. 2013 Journal of Ornithology, Becker et al. 2019 ICB), which posits that susceptibility to infection or intensity of parasite shedding would thus be greatest at the range margins (and therefore that is where humans would have the greatest risk of exposure to infectious hosts, in the spirit of Plowright et al. 2017’s Nature Reviews Microbiology framework on spillover hierarchies). I think the jump from hosts being at the range limits to spillover risk needs to be introduced in a much more mechanistic manner here. 

REPLY: We thank the reviewer for bringing this issue forward. We have now added a small paragraph where we discuss some of the relevant mechanisms that previous authors have discussed  (see lines 61-73).  We also come back to some of these mechanisms in the discussion (see lines 235-258).

L54: The disease is mis-spelled here (i.e., “Leischmaniasis” should be “leishmaniasis”).

REPLY: We have corrected this term (see line 54)

L57: Although not likely a directly transmitted parasite, the authors should also cite Briers 2003 “Range limits and parasite prevalence in a freshwater snail”, which provides a non-vector example of how parasite prevalence is greatest at the species range. 

REPLY: We thank the reviewer for this suggestion, and have now added the reference with a small statement to support it (see lines 57-58)

L60: See above comments on virus naming; please use the full virus name or just Marburg virus (and then MARV thereafter). 

REPLY: We have changed all instance to stick to the term “Marburg virus”

L60: This statement isn’t really accurate. MARV and EBOV are different genera of viruses. It would be more accurate to say “a filovirus related to the Ebola genera of viruses.”

REPLY: We have followed the reviewer’s suggestion, and reworded this statement (see line 76-77)

L68: The authors should also cite Amman et al. 2012 PLoS Pathogens here, which provides a direct test of shedding pulses in Rousettus bats being associated with spillover to humans.

REPLY: We thank the reviewer for this suggestion, and added the suggested references to our introduction (see lines 88-90)

L77-L88: See my above comment, but I think this introduction is really lacking information on the mechanistic jump from host occupancy in a range margin habitat to spillover. I think the authors need to better acknowledge that range limits are unlikely to affect spillover in a direct sense (e.g., reservoir-to-human contact) but rather affect more upstream processes (e.g., immunity, shedding) that then percolate down to spillover risk. In an ideal world, you would test of being at the range limit affects virus prevalence or shedding in the host, although I understand that isn’t possible with the dataset here. But I do think the authors should acknowledge that limitation here.

REPLY: We have now broadened the introduction to include the concepts of pathogens shedding and immunological responses.  We also now spend some time discussing potential directions for future work, including pointing out that studies of spatial variation in pathogen shedding could shed further light on the mechanisms at work (see lines 251-256)

L106-110: The authors should clarify here that the Rousettus distribution is patchy, such that it is composed of multiple polygons. In addition, is there any way to account for whether outbreaks are observed at the generally more northern or southern limit of the bat range? E.g., you could theoretically have an outbreak in very western Ethiopia that in this framework would correspond to the “edge” of one of the IUCN range polygons, but this is still closer to the core of the bat range than say an outbreak that would occur in South Africa or Turkey (very much at the southern or northern limits of the range).

REPLY: The reviewer makes a very valid point here. However, most of Marburg virus outbreaks happen within the same range of latitude. As such, they mostly all happen within the core of the Egyptian fruit bat distribution. We also feel that figure 1a adequately communicates to readers both that the range of the Egyptian fruit bat is patchy and a useful summary of where outbreaks have been documented.   The origin points of all outbreaks in Africa are depicted.  The only outbreaks that have occurred in any other region were in Europe, and occurred after contact with infected Green monkeys imported from this region (several outbreaks in Europe were started due to contact with monkeys from the same population in 1967).

L111: Please clarify the sample sizes used in the analyses here (i.e., is it N=13?).

REPLY: We have clarified the statement (see line 120)

L112-125: Since the authors use a regression framework to analyze outbreak size ~ range limit, why do you not do the same for the climatic predictors? It’s unclear why two separate analyses are used to analyze predictor A and predictor B. 

REPLY:  Detailed time series information is available for climate variables, which is why we chose a more nuanced moving window analyses for the climate data. Due to absence of any significant trends, we did not pursue further analyses such as outbreak size ~ climate. A formal regression analyses was conducted with range edge as a predictor, given that preliminary analyses and our map (Figure 1A) suggested a relationship between outbreak size and range edge.

L126: The authors mention using a multi-level model, but no random effects are apparent in the structure stated in L130 (which looks like outbreak size ~ range limit distance). Please also clarify the other Bayesian parameters for the brm() function (how many chains, how many iterations, how much burn in, etc). 

REPLY: We have added details on this at lines 140-141 and lines 146-147)

L131: Why was this particular geographic extent chosen? From Figure 1, it is clear that the authors do not predict on the whole Rousettus geographic range, just that within the African continent. Likewise, I do not see merit in predicting across the whole African continent if the Egyptian fruit bat is not distributed across the whole continent. Why not only predict within the known polygons? In general, I’m also pretty wary of predicting spillover risk across an entire continent based on just 13 data points.

REPLY: We agree with the reviewer that the predictions in figure 1 would have been misleading due to the small dataset. As such, we have removed the panel B, and replaced it with a figure showing model predictions of distance to range edge vs number of cases..

Figure 1: A legend should be provided in panel A to indicate outbreak size; can the point size also be increased and/or made transparent? From my understanding of the methods, there should be 13 outbreaks but I can only see 7-8 here. Also note my above comment, as both figures restrict the Egyptian fruit bat geographic range (this should be clarified in the legend). I would suggest using a better color scale for panels A and B (given concerns about red-green colorblindness). It is also very hard to make out the areas of high predicted human cases in panel B. Also see my above comment about predicting across the African continent, given that this bat species is not broadly distributed. 

REPLY: The reviewer is correct that only 8 points appear on the map in figure 1, this is due to multiple outbreaks happening at the same set of geographic coordinates. Please see the above reply for figure 1, Panel B.  We have also clarified this in the legend to Figure 1 (see lines 168-175).

L146: “Effect size” is a bit vague. Is this not just the posterior mean of the slope for distance to the range limit? It would be more appropriate to describe this as simply the slope/beta; I would also clarify that the CIs are credible intervals from brm(). 

REPLY: We now clarify that the effect size represents a slope, and we now state that we are using Bayesian credible intervals (see lines 189-190)

L174-192: See my above comments about the Introduction, the authors should discuss the (likely) possibility that range limits affect bat immunity in ways that facilitate greater intensity of infection and/or virus shedding. I’m not sure the ecotone argument really makes much sense unless there is compelling evidence that ecotones are more likely at the range limits. Since the range limits are shaped by climatic or other biophysical variables, I doubt you’d find range limits are also areas of greater human-wildlife interaction. Either way, I think the idea stress at range limits mentioned here necessitates some discussion of this alternative explanation. 

REPLY: We now include discussion on parasites shedding and immunological process that could shape the pattern we observe (see lines 251-256). See also lines 61-73 where we now discuss potential spill over mechanisms at species range limits, including ecotones, in more detail.

L202: The authors should also briefly discuss the caveat that many processes need to occur in-between the host being present in a range limit habitat to spillover occurring and that measures of viral prevalence or shedding in such habitats would be important data as well. 

REPLY: We have added this caveat to the discussion (see lines 257-272)

Round 2

Reviewer 1 Report

Authors have responded all the comments,

Author Response

We thank the reviewer for looking at this submission.

Reviewer 2 Report

The authors have done a nice job in their revision. I think there are a few relatively minor changes to make to this version:

1) I think the authors could be a bit more conservative with their language regarding the results in the Abstract given the small sample size (e.g., “Here, we test Schmalhausen’s law in a novel framework of Marburgvirus outbreaks originating from spillover events”). A more conservative phrasing might be “Here, we provide a preliminary test of Schmalhausen’s law using data on Marburg virus outbreaks originating from spillover events”). 

2) In the Abstract, also note that both uses of “Marburgvirus” (L15, L17) should be “Marburg virus”.

3) In L54, “Leishmaniasis” should be lower case.

4) In L57-58, the two sentences are contradictory (“There is some evidence that directly transmitted parasite prevalence is highest at the range edge” and then “This law has yet to be tested in a directly transmitted pathogen”)

5) L66-67: There is a very large body of literature on how local landscape features can affect immunity; I would remove “in some cases” here.

6) L69-73: I would still like to authors to acknowledge whether there is any support for range edges also showing more ecotones. I don’t think the Borremans piece explicitly covers whether ecotones are more likely at species range edges.

7) L152-153: Could the authors also provide the p value or 95% CI? Does this GLM use a frequentist or Bayesian approach? It’s unclear given that the subsequent analyses use brms.

8) L175: It is still unclear what random effects are in the brms model, given that the authors describe this as a multi-level model. 

9) Figure 1 is much improved. I still think the authors should reconsider the red/green color scheme given concerns about colorblind-friendly palettes. I would also suggest adding the raw data to panel B, especially given the text in L197 about the outlier (currently not visible). Please also clarify in the legend that the grey polygon in B is the (95%?) credible interval. 

10) L229: I would suggest the authors still re-cite the Briers 2003 study here and qualify the statement slightly (e.g., replace the second clause of the sentence here with “there are few tests of Schmalhausen’s law for directly transmitted pathogens”).

11) L249: I think this is another spot where language could be tempered a bit.

Author Response

The authors have done a nice job in their revision. I think there are a few relatively minor changes to make to this version:

REPLY: We thank the reviewer for taking the time to look at this manuscript again. Bellow, we present a detailed reply to the point raised by the reviewer

1) I think the authors could be a bit more conservative with their language regarding the results in the Abstract given the small sample size (e.g., “Here, we test Schmalhausen’s law in a novel framework of Marburgvirus outbreaks originating from spillover events”). A more conservative phrasing might be “Here, we provide a preliminary test of Schmalhausen’s law using data on Marburg virus outbreaks originating from spillover events”). 

REPLY: We have made the suggested change (see lines 14-15)

2) In the Abstract, also note that both uses of “Marburgvirus” (L15, L17) should be “Marburg virus”.

REPLY: We have made the suggested change (see line 15)

3) In L54, “Leishmaniasis” should be lower case.

REPLY: We have made the suggested change (see line 56)

4) In L57-58, the two sentences are contradictory (“There is some evidence that directly transmitted parasite prevalence is highest at the range edge” and then “This law has yet to be tested in a directly transmitted pathogen”)

REPLY: We agree with the reviewer that this sentence might appear contradictory. However, Biers (2003) does not directly test for the influence of Schmalhausen’s law on outbreak characteristic per se.  We have edited the last sentence in this paragraph for clarity (see lines 60-61)

5) L66-67: There is a very large body of literature on how local landscape features can affect immunity; I would remove “in some cases” here.

REPLY: We have made the suggested change

6) L69-73: I would still like to authors to acknowledge whether there is any support for range edges also showing more ecotones. I don’t think the Borremans piece explicitly covers whether ecotones are more likely at species range edges.

REPLY: We agree with the reviewer that species edge do not always occur at the same place as ecotones (see Lourenço et al. 2019). However, we mention in this statement that, only when both conditions are met (e.g. presence of ecotone and species edge), increased in pathogen shedding by natural host is likely to translate in spillover events.  We have modified this sentence slightly to make it more clear to readers that this is a pattern that may occur in some instances, rather than an expected universal pattern (see line 61).   

7) L152-153: Could the authors also provide the p value or 95% CI? Does this GLM use a frequentist or Bayesian approach? It’s unclear given that the subsequent analyses use brms.

REPLY: We have made the suggested change (see lines 146-150)

8) L175: It is still unclear what random effects are in the brms model, given that the authors describe this as a multi-level model. 

REPLY: We have reworded this statement (see line 167)

9) Figure 1 is much improved. I still think the authors should reconsider the red/green color scheme given concerns about colorblind-friendly palettes. I would also suggest adding the raw data to panel B, especially given the text in L197 about the outlier (currently not visible). Please also clarify in the legend that the grey polygon in B is the (95%?) credible interval. 

REPLY: We have made the suggested changes, and used a colour-blind palette for figure 1, panel A (See figure 1 and legend)

10) L229: I would suggest the authors still re-cite the Briers 2003 study here and qualify the statement slightly (e.g., replace the second clause of the sentence here with “there are few tests of Schmalhausen’s law for directly transmitted pathogens”).

REPLY: We have added a reference to Briers (2003) at the beginning of this section (see line 232).  While Briers did not set out to test Schmalhausen’s law, they did demonstrate a pattern predicted by it.

11) L249: I think this is another spot where language could be tempered a bit.

REPLY: We have tempered the language of this statement (see lines 254 and 259)
